

# Morphometric variation of extant platyrrhine molars: taxonomic implications for fossil platyrrhines

Mónica Nova Delgado[1], Jordi Galbany[1,2] and Alejandro Pérez-Pérez[1]

[1] Department of Evolutionary Biology, Ecology and Environmental Sciences, Zoology and Physical Anthropology Section, Universitat de Barcelona, Barcelona, Spain
[2] Center for the Advanced Study of Human Paleobiology, Department of Anthropology, George Washington University, Washington DC, United States of America

## ABSTRACT

The phylogenetic position of many fossil platyrrhines with respect to extant ones is not yet clear. Two main hypotheses have been proposed: the *layered* or *successive radiations* hypothesis suggests that Patagonian fossils are Middle Miocene stem platyrrhines lacking modern descendants, whereas the *long lineage* hypothesis argues for an evolutionary continuity of all fossil platyrrhines with the extant ones. Our geometric morphometric analysis of a 15 landmark-based configuration of platyrrhines' first and second lower molars suggest that morphological stasis may explain the reduced molar shape variation observed. Platyrrhine lower molar shape might be a primitive retention of the ancestral state affected by strong ecological constraints throughout the radiation of the main platyrrhine families. The Patagonian fossil specimens showed two distinct morphological patterns of lower molars, *Callicebus*—like and *Saguinus*—like, which might be the precursors of the extant forms, whereas the Middle Miocene specimens, though showing morphological resemblances with the Patagonian fossils, also displayed new, derived molar patterns, *Alouatta*—like and *Pitheciinae*—like, thereby suggesting that despite the overall morphological stasis of molars, phenotypic diversification of molar shape was already settled during the Middle Miocene.

## INTRODUCTION

Platyrrhine evolution is controversial. However, most researchers agree that they most likely constitute a monophyletic clade derived from African ancestors (*Fleagle & Kay, 1997*; *Takai et al., 2000*; *Kay et al., 2004*; *Oliveira, Molina & Marroig, 2009*; *Bond et al., 2015*), although the phylogenetic position of some living taxa and the affinities of some fossil specimens are still uncertain. Currently, two different viewpoints have been proposed regarding the evolutionary history of the earliest platyrrhines and their overall relationships with extant forms. The "long lineages" hypothesis argues that the oldest known Patagonian fossils (16–20 Ma) are to be included within the extant Platyrrhines (*Rosenberger, 1979*; *Rosenberger, 1980*; *Rosenberger, 1981*; *Rosenberger, 1984*; *Rosenberger et al., 2009*; *Tejedor, 2013*), whereas the "layered or successive radiations" hypothesis suggests that these fossils constitute a geographically isolated stem group, phylogenetically unrelated to the crown platyrrhines,

Corresponding author
Alejandro Pérez-Pérez,
martinez.perez-perez@ub.edu

that went extinct (along with some Antillean species) (*Kay, 2010*; *Kay, 2014*; *Kay & Fleagle, 2010*; *Kay et al., 2008*). According to *Kay (2014)*, the divergence of modern lineages occurred in the tropics. The Late Oligocene and Early Miocene platyrrhines would have branched off from the ancestral lineage when climatic conditions in Patagonia became unfavorable and the Andean uplift was a potential barrier to their dispersal. However, *Tejedor (2013)* has suggested that *Chilecebus* (20 Ma), a fossil specimen (*Tejedor, 2003*) from the western Andean cordillera, south of Santiago de Chile, indicates that the Andean mountains did not constitute a biogeographic barrier. *Tejedor (2013)* argued that a paleobiogeographic corridor throughout western South America would have allowed for a continental connectivity between the north and the southernmost fossil platyrrhines. Unfortunately, dating of the fossil specimens and fossil-based approaches for calibrating the molecular phylogeny support both models. *Perez et al. (2013)* have estimated a crown platyrrhine origin at around 29 Ma (27–31), which allows for the inclusion of the fossil Patagonian primates into a crown Platyrrhini lineage showing evolutionary continuity with the Middle Miocene lineages. In contrast, *Hodgson et al. (2009)* have dated their origin between 16.8 and 23.4 Ma, suggesting an unlikely relationship of the early Miocene fossils with the crown platyrrhine clade (but see different temporal models in *Goodman et al., 1998*; *Opazo et al., 2006*; *Chatterjee et al., 2009*; *Perelman et al., 2011*; *Wilkinson et al., 2011*; *Jameson Kiesling et al., 2014*).

Molar morphology has been widely used to determine the phylogenetic positions of extinct specimens with respect to living forms (e.g., *Kay, 1990*; *Rosenberger et al., 1991*; *Rosenberger, Setoguchi & Hartwig, 1991*; *Benefit, 1993*; *Meldrum & Kay, 1997*; *Miller & Simons, 1997*; *Horovitz & MacPhee, 1999*; *Kay & Cozzuol, 2006*; *Kay et al., 2008*), since tooth development is under strong genetic control (*Jernvall & Jung, 2000*). Recent studies have reported that molar shapes carries strong phylogenetic signals, and can be a useful tool for establishing taxonomic affinities between extant and extinct catarrhine primates (*Nova Delgado et al., 2015a*; *Gamarra et al., 2016*), and also in some Platyrrine taxa (*Nova Delgado et al., 2015b*) with closely related species exhibiting common phenotypic traits.

## Affinities of the fossil platyrrhine primates based on dental morphology

Until now, a total of 31 Early Miocene Platyrrhini fossil genera have been reported in the South American continent and the Caribbean: 13 in La Venta (Colombia), eight in the Argentinian Patagonia, five in the Greater Antilles, five in Brazil, and one each in Chile, Bolivia and Peru (*Tejedor, 2013*; *Bond et al., 2015*). *Neosaimiri*, *Laventiana* (La Venta, Colombia) and *Dolichocebus* (Chubut Province, Argentina) have been included in Cebinae (*Rosenberger, 2011*). *Neosaimiri* is considered a direct ancestor of the extant *Saimiri* due to its similar molar shape (*Rosenberger, Setoguchi & Shigehara, 1990*; *Rosenberger et al., 1991*). Its molars exhibit sharp cusps, well-developed distal cusps, buccal cingulum, a strong buccal flare, and a distinct post-entoconid notch on molars only found in *Saimiri* and *Laventiana* (*Rosenberger et al., 1991*; *Rosenberger, Setoguchi & Hartwig, 1991*; *Takai, 1994*; *Tejedor, 2008*). *Laventiana* is sometimes considered a synonym of *Neosaimiri* (*Takai, 1994*; *Meldrum & Kay, 1997*), although it has been suggested to be more primitive than

*Neosaimiri* (*Rosenberger, Setoguchi & Hartwig, 1991*). *Laventiana*'s teeth closely resemble those of *Saimiri* and *Cebus-Sapajus*; it shows thick-enamel bunodont molars exhibiting a small buccal cingulum and an angular cristid obliqua, lacking buccal flare (*Rosenberger, Setoguchi & Hartwig, 1991*). *Dolichocebus* has been suggested to be a member of the *Saimiri* lineage, mainly for its interorbital fenestra considered a derived feature in squirrel monkeys (*Tejedor, 2008*; *Rosenberger et al., 2009*; *Rosenberger, 2010*). However, Kay and colleagues (*Kay et al., 2008*; *Kay & Fleagle, 2010*) argued that *Dolichocebus* is a stem platyrrhine and that the description of the orbital region was probably affected by postmortem damage.

On the other hand, *Aotus dindensis* was first described as a sister taxon of extant *Aotus* (*Setoguchi & Rosenberger, 1987*), although *Kay (1990)* has suggested that it is probably conspecific with *Mohanamico hershkovitzi*, which may be closely related to the callitrichines, especially *Callimico*, due to their morphological similarities in the canine and the second premolar. *Aotus dindensis* is included into the Pitheciidae (*Rosenberger, Setoguchi & Shigehara, 1990*) within the Homunculinae subfamily, along with *Aotus, Callicebus* and some Argentinian and Caribbean fossil primates (*Rosenberger, 1981*; *Rosenberger, 2002*; *Rosenberger, 2011*). However, molecular phylogenetic analyses have repeatedly rejected a link between *Aotus* and Pitheciids (e.g., *Hodgson et al., 2009*; *Osterholz, Walter & Roos, 2009*; *Wildman et al., 2009*), placing it as a basal cebid. *Tejedor & Rosenberger (2008)* proposed that *Homunculus* is likely an ancestral pitheciid because although it shows a primitive dental morphology, it notably resembles that of *Callicebus*. The two taxa show rectangular-shaped molars, small incisors and non-projecting canines, a trait shared with *Carlocebus* (*Fleagle, 1990*). Nonetheless, unlike *Callicebus*, the molars of *Homunculus* exhibit well-marked crests and prominent cusps (*Tejedor, 2013*), and an unusual paraconid on the lower first molar (also found in *Dolichocebus*; *Kay et al., 2008*). Another fossil from the early Miocene known as *Soriacebus* was initially included as an early Pitheciinae (*Rosenberger, Setoguchi & Shigehara, 1990*), due to its resemblance on the anterior dentition (*Fleagle et al., 1987*; *Fleagle, 1990*; *Fleagle & Tejedor, 2002*; *Tejedor, 2005*). However, some dental traits of *Soriacebus* (premolars-molars size, lower molar trigonid, and reduced hypocone) bear resemblance also with the callitrichines. Indeed, *Kay (1990)* argues that such similarities found between *Soriacebus* and pitheciines or with callitrichines are due to homoplasy, rather than phylogenetic relationships among such lineages (*Kay, 1990*). According to *Kay (1990)*, *Soriacebus, Carlocebus, Homunculus* and all Patagonian fossils should be considered stem platyrrhines.

*Xenothrix* is a Late Pleistocene Caribbean fossil from Jamaica that shows a callitrichine-like dental formula (2132; *MacPhee & Horovitz, 2004*), low relief molars and a narrowing of intercuspal distance and augmentation of the mesial and distal crown breadths (*Cooke, Rosenberger & Turvey, 2011*), a feature also seen in *Insulacebus toussaintiana*, another Caribbean primate. *Rosenberger (2002)* argued that *Xenothrix* is closely related to *Aotus* and *Tremacebus* by the enlargement of the orbits and the central incisors, while *MacPhee & Horovitz (2004)* suggested a possible Pitheciidae affinity, due to its low relief molar pattern. Nonetheless, the puffed cusps and the lack of crenulation on the molar crown discriminate the Jamaican fossil from the Pitheciidae, suggesting that it is likely that *Xenothrix* does not belong to crown platyrrhine group (*Kay, 1990*; *Kinzey, 1992*).

*Cebupithecia* and *Nuciruptor*, two Colombian Middle Miocene genera, also share some traits with the extant Pitheciidae family, mostly in the anterior dentition but also in their low molar cusps and poorly developed crests (*Kay, 1990*; *Meldrum & Kay, 1997*). *Nuciruptor* does not exhibit several of the shared traits among pitheciines (projecting canine and small or absent diastema). *Cebupithecia*, although considered to be more derived than *Nuciruptor* (*Meldrum & Kay, 1997*), was interpreted by *Meldrum & Kay (1997)* as an example of convergent evolution, and thus, not a direct ancestor of extant pitheciines. Finally, *Stirtonia* (originally from Colombia but also recovered from Acre State, Brazil) exhibits similar dental size and morphology to extant *Alouatta*; showing molar teeth with sharp and well-formed crests, a long cristid oblique, small trigonid, and spacious talonid basin (*Hershkovitz, 1970*; *Kay et al., 1987*; *Kay & Frailey, 1993*; *Kay & Cozzuol, 2006*; *Kay, 2014*).

Numerous studies have examined landmark-based geometric morphometrics (GM) of molar shape for studying patterns of inter-specific variation and their implication in phylogeny and ecological adaptations (e.g., *Bailey, 2004*; *Cooke, 2011*; *Gómez-Robles et al., 2007*; *Gómez-Robles et al., 2008*; *Gómez-Robles et al., 2011*; *Martinón-Torres et al., 2006*; *Singleton et al., 2011*; *Nova Delgado et al., 2015a*; *Nova Delgado et al., 2015b*; *Gamarra et al., 2016*). However, in Platyrrhine primates, GM of molar shape has mainly focused on dietary adaptations (*Cooke, 2011*), rather than to predict the phylogenetic attribution of unclassified specimens (*Nova Delgado et al., 2015a*).

The aim of the present study is to use two-dimensional (2D) GM to quantify and analyze occulsal shape variation of lower molars ($M_1$ and $M_2$) of extant Platyrrhini primates to asesses the affinities of the Patagonian, Colombian, and Antillanean fossil taxa with the extant forms and to estimating the efficiency of molar shape for discriminating fossil specimens.

## MATERIAL AND METHODS

Images of the dental crowns, in occlusal view and including a scale line, of 12 holotype fossil platyrrhine specimens and one fossil from Fayum (*Proteopithecus sylviae*), were obtained from the literature (Table 2). The platyrrhine fossil specimens included 12 genera (*Soriacebus*, *Dolichocebus*, *Homunculus*, *Carlocebus*, *Neosaimiri*, *Laventiana*, *Mohanamico*, *Aotus*, *Stirtonia*, *Nuciruptor*, *Cebupithecia*, and *Xenothrix*), discovered in Argentina, Colombia, and Jamaica, and dated to between Holocene and early Miocene (Table 1).

The extant comparative samples consisted in 802 adult individuals representing all recognized platyrrhine groups (three families, 18 genera, 61 species; Table 2), whose 2D and 3D morphometric variability of lower molars has alredy been analysed in some platyrrine species (*Nova Delgado et al., 2015b*). Dental casts were obtained from original specimens housed at Museu de Zoologia Universidade de São Paulo (MZPS), Museu Nacional do Rio de Janeiro (MNRJ) in Brazil, and from Hacienda La Pacífica (HLP) in Costa Rica. The casts were made following published protocols (see *Galbany, Martínez & Pérez-Pérez, 2004*; *Galbany et al., 2006*). 2D images of molar occlusal surfaces of the extant specimens were taken with a Nikon D70 digital camera fitted with a 60-mm optical lens held horizontally on the stand base, at a minimum distance of 50 cm. The dental crown

**Table 1  List of fossils used in the study.**

| Fossils | Location | Age (Ma) | Phylogenetic position | Specimen number and reference |
|---|---|---|---|---|
| **F1** *Proteopithecus sylviae* | Fayum, Egypt | 33.9–28.4[a] | Stem anthropoid[b] | CGM 42209; *Miller & Simons (1997)* |
| **F2** *Soriacebus* spp. | Pinturas Formation, Santa Cruz Province, Argentina | 17[c] | Stem platyrrhine[d]/Pitheciidae[e] | MACN-SC 2[1], MACN-SC 5[2] MPM-PV 36[3]; *Tejedor (2005)* |
| **F3** *Dolichocebus gaimanesis* | Gaiman, Chubut Province, Argentina | 20[f] | Stem platyrhine/sister to *Saimiri*[g] | MPEF 5146; *Kay et al. (2008)* |
| **F4** *Homunculus* spp. | Santa Cruz Formation, Santa Cruz Province, Argentina | 16.5[h] | Stem platyrrhine/Pitheciidae | MACN-A5969; *Tejedor & Rosenberger (2008)* |
| **F5** *Carlocebus* spp. | Pinturas Formation, Santa Cruz Province, Argentina | 18–19[i] | Stem platyrrhine/Pitheciidae | MACN-SC 266; *Fleagle (1990)* |
| **F6** *Neosaimiri fieldsi* | La Venta, Huila, Colombia | 13.5–11.8[j] | Sister to *Saimiri*[k] | IGM-KU 89029[4], IGM-KU 89019[5], UCMP 39205[6], IGM-KU 89002[7], IGM-KU 39034[8], IGM-KU 89053[9], IGM-KU 89130[10]; *Takai (1994)* |
| **F7** *Laventiana annectens* | La Venta, Huila, Colombia | 13.5–11.8 | Sister to *Saimiri*/synonymy with *Neosaimiri*[l] | IGM-KU 880; *Rosenberger, Setoguchi & Hartwig (1991)* |
| **F8** *Mohanamico hershkouitzi* | La Venta, Huila, Colombia | 13.5–11.8 | Sister to *Callimico*[m] | IGM 181500; *Kay (1990)* |
| **F9** *Aotus dindensis* | La Venta, Huila, Colombia | 13.5–11.8 | Sister to *Aotus*[n]/coespecific with *Mohanamico*[o] | IGM-KU 8601; *Kay (1990)* |
| **F10** *Stirtonia* spp. | La Venta, Huila, Colombia | 13.5–11.8 | sister to *Alouatta*[p] | UCPM 38989; *Kay et al. (1987)* |
| **F11** *Nuciruptor rubricae* | La Venta, Huila, Colombia | 13.5–11.8 | Pitheciidae[q]/stem Pitheciinae[r] | IGM 251074; *Meldrum & Kay (1997)* |
| **F12** *Cebupithecia sarmientoni* | La Venta, Huila, Colombia | 13.5–11.8 | Pitheciidae/stem Pitheciinae | UCMP 38762; *Meldrum & Kay (1997)* |
| **F13** *Xenothrix macgregori* | Jamaica | | Holocene[s] stem platyrhine/retaded to *Callicebus*[t] | AMNHM 148198; *MacPhee & Horovitz (2004)* |

**Notes.**

References used in the table: (*Miller & Simons, 1997*)[a]; (*Kay, 1990*)[b]; (*Fleagle et al., 1987*)[c]; (*Kay, 2010*; *Kay, 2014*[f]; *Kay & Fleagle, 2010*; *Kay et al., 2008*[f]); (*Rosenberger, 1979*)[g]; *Tejedor & Rosenberger, 2008*[h]); (*Rosenberger, 1979*)[g]; (*Fleagle, 1990*)[i]; (*Flynn, Guerrero & Swisher, 1997*)[j]; (*Rosenberger, Setoguchi & Hartwig, 1991*)[k]; (*Takai, 1994*; *Meldrum & Kay, 1997*)[l]; (*Rosenberger, Setoguchi & Shigehara, 1990*)[m]; (*Setoguchi & Rosenberger, 1987*; *Takai et al., 2009*)[n]; *Meldrum & Kay, 1997*[o,q]; (e.g., *Hershkovitz, 1970*; *Kay et al., 1987*)[p]; (*Cooke, Rosenberger & Turvey, 2011*)[s]; (*MacPhee & Horovitz, 2004*)[t].

Institutional abbreviations: CGM, Cairo Geological Museum; MPM-PV, Museo Regional Provincial Padre Manuel Jesús Molina, Río Gallegos, Argentina; MPEF, Museo Paleontológico E. Feruglio, Trelew, Chubut Province, Argentina; MACN, MACN-SC/A, Museo Argentino de Ciencias Naturales "Bernardino Rivadavia," Buenos Aires, Argentina; SC/A, denotes locality; IGM, IGM-KU, Museo Geologico del Instituto Nacional de Investigaciones Geológico-Mineras, Bogota, Colombia; KU, denotes Kyoto University; UCPM, University of California Museum of Paleontology, Berkeley, California; AMNHM, Division of Vertebrate Zoology Mammalogy, American Museum of Natural History.

was imaged with a 0° of tilt with the cervical line perpendicular to the camera focus (*Nova Delgado et al., 2015a*). Images of fossil dental crowns were obtained from the literature and imported to Adobe Photoshop, where they were scaled to the same resolution (400 dpi). The images both for the extant and the fossil specimens were scaled to 5 mm and standardized to right side, with the mesial border facing to the right, the distal border to the left, and the lingual and buccal sides facing upward and downward, respectively. All images were saved at high resolution (1600 × 1200 pixel) in JPEG format.

**Table 2 List of the specimens included in this analysis of $M_1$ and $M_2$.** The subfamily-level classification was proposed by *Groves (2005)*.

| Genus/species | $M_{1-2}$ | Collection[a] |
|---|---|---|
| **Subfamily: Cebinae** | | |
| ***Cebus*** (gracile capuchins) | | |
| **1** *C. albifrons* | 9 | MZUSP, MNRJ |
| **2** *C. olivaceus* | 6 | MNRJ |
| ***Sapajus*** (robust capuchins) | | |
| **3** *S. apella* | 14 | MZUSP |
| **4** *S. libidinosus* | 15 | MNRJ |
| **5** *S. nigritus* | 15 | MNRJ |
| **6** *S. robustus* | 15 | MNRJ |
| **7** *S. xanthosternos* | 7 | MNRJ |
| **Subfamily: Samiriinae** | | |
| ***Saimiri*** (squirrel monkeys) | | |
| **8** *S. boliviensis* | 17 | MZUSP, MNRJ |
| **9** *S. sciureus* | 25 | MZUSP, MNRJ |
| **10** *S. ustus* | 18 | MZUSP, MNRJ |
| **11** *S. vanzolinii* | 8 | MNRJ |
| **Subfamily: Callitrichinae** | | |
| ***Callithrix*** (marmosets from Atlantic Forest) | | |
| **12** *C. aurita* | 11 | MNRJ |
| **13** *C. geoffroyi* | 15 | MNRJ |
| **14** *C. jacchus* | 21 | MZUSP |
| **15** *C. kuhlii* | 20 | MNRJ |
| **16** *C. penicillata* | 14 | MNRJ |
| ***Mico*** (marmosets from Amazon) | | |
| **17** *M. argentata* | 21 | MZUSP, MNRJ |
| **18** *M. chrysoleuca* | 16 | MZUSP, MNRJ |
| **19** *M. emiliae* | 6 | MZUSP |
| **20** *M. humeralifer* | 16 | MZUSP |
| **21** *M. melanurus* | 8 | MZUSP, MNRJ |
| ***Cebuella*** (pygmy marmoset) | | |
| **22** *C. pygmaea* | 7 | MZUSP |
| ***Callimico*** (goeldi's marmoset) | | |
| **23** *C. goeldii* | 4 | MZUSP |
| ***Leontopithecus*** (lion tamarins) | | |
| **24** *L. chrysomelas* | 5 | MZUSP, MNRJ |
| **25** *L. rosalia* | 17 | MZUSP, MNRJ |
| ***Saguinus*** (tamarins) | | |
| **26** *S. fuscicollis* | 13 | MZUSP |
| **27** *S. imperator* | 10 | MZUSP |
| **28** *S. labiatus* | 9 | MZUSP, MNRJ |

**Table 2** (*continued*)

| Genus/species | $M_{1-2}$ | Collection[a] |
|---|---|---|
| **29** *S. midas* | 22 | MZUSP, MNRJ |
| **30** *S. mystax* | 13 | MZUSP, MNRJ |
| **31** *S. niger* | 14 | MNRJ |
| **Subfamily: Aotinae** | | |
| ***Aotus*** (owl or night monkeys) | | |
| **31** *A. azarae* | 4 | MZUSP, MNRJ |
| **32** *A. nigriceps* | 9 | MZUSP, MNRJ |
| **33** *A. trivirgatus* | 21 | MZUSP |
| **Subfamily: Callicebinae** | | |
| ***Callicebus*** (titi monkeys) | | |
| **34** *C. bernhardi* | 5 | MNRJ |
| **35** *C. cupreus* | 14 | MZUSP, MNRJ |
| **36** *C. hoffmannsi* | 12 | MNRJ |
| **37** *C. moloch* | 16 | MZUSP, MNRJ |
| **38** *C. nigrifrons* | 8 | MNRJ |
| **39** *C. personatus* | 16 | MZUSP, MNRJ |
| **Subfamily: Pitheciinae** | | |
| ***Cacajao*** (uakaris) | | |
| **40** *C. calvus* | 14 | MZUSP, MNRJ |
| **41** *C. melanocephalus* | 9 | MZUSP, MNRJ |
| ***Chiropotes*** (bearded sakis) | | |
| **42** *C. albinasus* | 18 | MZUSP, MNRJ |
| **43** *C. satanas* | 15 | MZUSP, MNRJ |
| ***Pithecia*** (sakis) | | |
| **44** *P. irrorata* | 17 | MZUSP, MNRJ |
| **45** *P. monachus* | 7 | MZUSP, MNRJ |
| **46** *P. pithecia* | 16 | MZUSP, MNRJ |
| **Subfamily: Atelinae** | | |
| ***Lagothrix*** (woolly monkeys) | | |
| **47** *L. cana* | 7 | MNRJ |
| **48** *L. lagotricha* | 8 | MZUSP |
| ***Brachyteles*** (muriquis) | | |
| **49** *B. arachoides* | 16 | MZUSP, MNRJ |
| **50** *B. hypoxanthus* | 5 | MNRJ |
| ***Ateles*** spider monkeys) | | |
| **51** *A. belzebuth* | 2 | RBINS |
| **52** *A. chamek* | 15 | MNRJ |
| **53** *A. marginatus* | 20 | MZUSP |
| **Subfamily: Alouatinae** | | |
| ***Alouatta*** (howler monkeys) | | |
| **54** *A. belzebul* | 15 | MZUSP |
| **55** *A. caraya* | 15 | MZUSP, MNRJ |

**Table 2** (*continued*)

| Genus/species | $M_{1-2}$ | Collection[a] |
|---|---|---|
| **56** *A. discolor* | 10 | MNRJ |
| **57** *A. guariba* | 5 | MZUSP, MNRJ |
| **58** *A. g. clamitas*[†] | 15 | MNRJ |
| **59** *A. nigerrima* | 10 | MNRJ |
| **60** *A. palliata* | 15 | HLP |
| **61** *A. seniculus* | 15 | MZUSP |
| **62** *A. ululata* | 7 | MNRJ |

**Notes.**
[†] Subspecies of *Alouatta guariba*
[a] Institutional abbreviations: MZUSP, Museu de Zoologia Universidade de São Paulo (Brazil); MNRJ, Museu Nacional do Rio de Janeiro (Brazil); HLP, Hacienda La Pacífica.

## Geometric morphometric analysis

Geometric Morphometrics (GM) quantifies shape differences between biological structures using a set of digitized homologous points (landmarks) in two-dimensional or three-dimensional spaces (*Bookstein, 1991*; *Adams, Rohlf & Slice, 2004*; *Slice, 2005*). Landmarks are numerical values (coordinates) that reflect the location and orientation of each specimen in the morphospace (*Slice, 2007*). A previously defined two-dimensional (2D) landmark protocol (*Nova Delgado et al., 2015a*; *Nova Delgado et al., 2015b*; *Gamarra et al., 2016*) was adopted. The configuration consisted of 15 landmarks. Molar occlusal polygon was defined by the tips of the four main cusps (protoconid, metaconid, hypoconid, and entoconid). The crown outline was represented by eight landmarks, which included two landmarks on fissure intersections; four corresponding to maximum crown curvatures; and two in the mid mesio-distal line on the crown perimeter. Furthermore, three landmarks were used to represent the positions of crests (Table 3 and Fig. 1) (*Cooke, 2011*). Landmark recording was performed with TPSDig v 1.40 (*Rohlf, 2004*) and landmark coordinates were then imported into MorphoJ (*Klingenberg, 2011*). The most commonly employed method to remove the information unrelated to shape variation is the generalized procrustes analysis (GPA) (*Rohlf, 1999*; *Rohlf, 2005*). GPA is based on a least squares superimposition approach that involves scaling, translation and rotation effect so that the distances between the corresponding landmarks are minimized (*Rohlf & Slice, 1990*; *Goodall, 1991*; *Rohlf & Marcus, 1993*; *Rohlf, 1999*; *Adams, Rohlf & Slice, 2004*).

Intra-observer landmark digitizing error was measured in a subsample of five specimens representative of five different species including one fossil taxon (*Alouatta belzebul*, *Aotus dindensis*, *Callicebus personatus*, *Callithrix geoffroyi*, *Pithecia irrorata*). The landmarks were digitized nine times during three non-consecutive days. Mean Procrustes distance between paired repetitions was 0.13328, with a standard deviation of 0.04644, and the average Pearson correlation between Procrustes distance matrices (Mantel test) of the repeated measurements was 0.9887. No significant differences in shape configurations among repetitions were obtained with a non-parametric MANOVA test (*Anderson, 2001*); $F = 0.07729$; $P = 0.9997$. The inter-observed error rates were not computed since a single researcher (MND) made all the analyses.

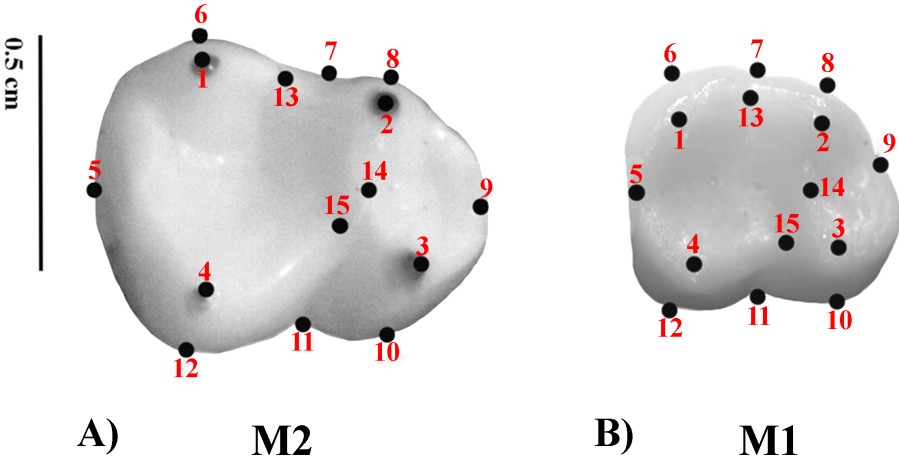

**Figure 1** **Set of landmarks used in the geometric morphometrics analyses.** (A) $M_2$; *Alouatta guariba* 23177 MNRJ; (B) $M_1$: *Sapajus libidinosus* 23246 MNRJ.

**Table 3** **Landmarks considered for the geometric morphometrics analysis of dental crown shape.**

| Landmark | Type | Definition |
|---|---|---|
| 1 | 2 | Tip of the distolingual cusp (entoconid) |
| 2 | 2 | Tip of the mesiolingual cusp (metaconid) |
| 3 | 2 | Tip of the mesiobuccal cusp (protoconid) |
| 4 | 2 | Tip of the distobuccal cusp (hypoconid) |
| 5 | 3 | Most distal point of the mid mesiodistal line on the crown outline |
| 6 | 2 | Point of maximum curvature directly below the entoconid[a] |
| 7 | 3 | Point on the dental crown outline at the lingual groove |
| 8 | 2 | Point of maximum curvature directly below the metaconid[a] |
| 9 | 3 | Most mesial point of the mid mesiodistal line on the crown outline |
| 10 | 2 | Point of maximum curvature directly below the protoconid[a] |
| 11 | 3 | Point on the dental crown outline at the mesial groove |
| 12 | 2 | Point of maximum curvature directly below the hypoconid[a] |
| 13 | 2 | Midpoint between the preentocristid and postmetacristid[a] |
| 14 | 2 | Lowest point on the protocristid[a] |
| 15 | 2 | Lowest point on the crista oblique[a] |

**Notes.**
[a]Landmarks follow definitions by *Cooke (2011)*.

After the procrustes superimposition, the covariance matrix of all the compared shapes was used to derive a Principal Components Analysis (PCA) (*Zelditch et al., 2004*). The PCA of $M_1$ and $M_2$ morphometric variability of the extant species were used to explore phenetic dental similarities of fossil specimens within the extant comparative platyrrhine sample. The resulting PCA scores were used to conduct a Linear Discriminant Function analysis (LDA) to classify fossil specimens, since PCA removes the irrelevant and redundant dimensions (*Zelditch et al., 2004*). LDA maximizes differences between groups but allows classifying isolated cases based on their distances to the group centroids of the extant

**Table 4 A comparison of distinct platyrrhine classifications at the subfamily level.**

| Genus | Subfamily by *Groves (2005)* | Subfamily by *Rosenberger (2011)* |
|---|---|---|
| *Cebus* | Cebinae | Cebinae |
| *Sapajus* | | |
| *Saimiri* | Saimiriinae | |
| *Callithrix* | Callitrichinae | Callitrichinae |
| *Mico* | | |
| *Cebuella* | | |
| *Callimico* | | |
| *Leontopithecus* | | |
| *Saguinus* | | |
| *Aotus* | Aotinae | Homunculinae |
| *Callicebus* | Callicebinae | |
| *Cacajao* | Pitheciinae | Pitheciinae |
| *Chiropotes* | | |
| *Pithecia* | | |
| *Lagothrix* | Atelinae | Atelinae |
| *Brachyteles* | | |
| *Ateles* | | |
| *Alouatta* | Alouattinae | |

taxa. The probability that a case belongs to a particular group is proportional to the distance to the group centroid (*Kovarovic et al., 2011*). The reliability of the classification was estimated from the *post-hoc* correct classification probability after cross-validation (*pcc*), and the *a posteriori* probability score was used as the probability that a fossil belongs to a particular group. Several LDAs were made considering different discriminant factors: (1) family (Cebidae, Atelidae, Pitheciidae), (2) the subfamily-level classification proposed by *Groves (2005)* (Subfamily G) (Cebinae, Saimiriinae, Callitrichinae, Pitheciinae, Callicebinae, Aotinae, Atelinae, Alouattinae), (3) the subfamily classification by *Rosenberger (2011)* (Subfamily R) (Cebinae, Callitrichinae, Pitheciinae, Homunculinae, Atelinae) (Table 4), and (4) a genus level (*Cebus, Sapajus, Saimiri, Callithrix, Mico, Cebuella, Callimico, Leontopithecus, Saguinus, Aotus, Callicebus, Cacajao, Chiropotes, Pithecis, Lagothrix, Brachyteles, Atelles, Allouatta*). The LDA analyses were carried out with SPSS v.15 (SPSS, Chicago, IL, USA).

## RESULTS

### Principal components analyses

The first two PCs of the PCA analysis of $M_1$ for all platyrrhines (Fig. 2) explain 42.06 % of total shape variance (PC1 30.60%; PC2 11.46%). Positive scores on PC1 correspond to molars with a broad occlusal polygons and a mesiodistally rectangular outline; whereas negative PC1 scores characterize a relatively quadrangular outline and slight buccolingually rectangular occlusal polygon resulted by displacement of distal cusps (entoconid and

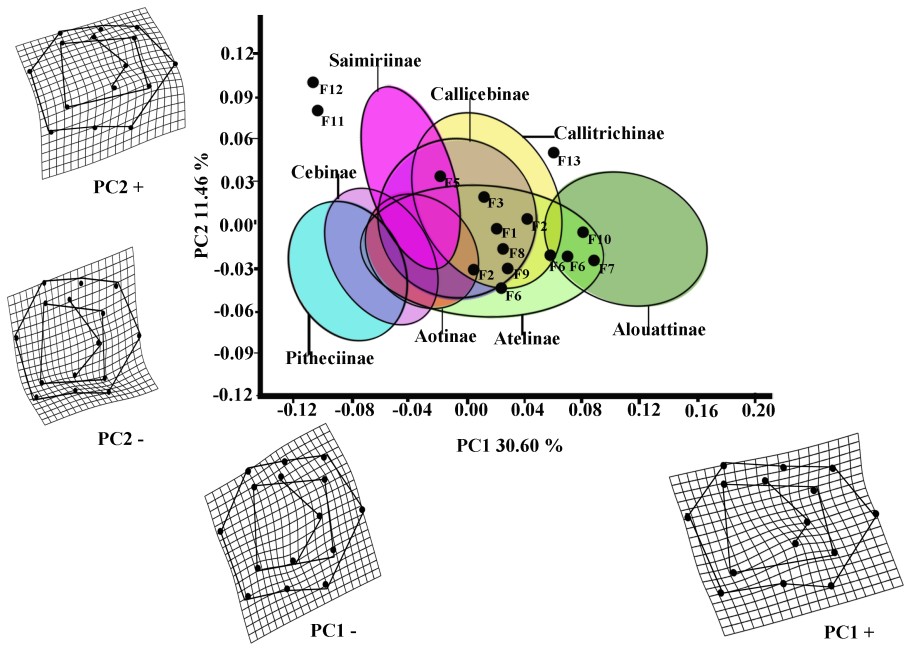

**Figure 2** **Scatterplot of the first two principal components (PCs) derived from the PCA of $M_1$ shape variability of Platyrrhini.** Grids indicate the deformations associated with the extreme values of each principal component. Ellipses represent the subfamily-level classification proposed by *Groves (2005)*. The letters F and numbers in figure represent the fossils listed in Table 1.

hypoconid) to mesio-lingually and mesial cusps (metaconid and protoconid) to distal-lingually side respectively. Positive scores on PC2 molar indicate a rectangular occlusal polygon and a mesiodistally rectangular outline, whereas negative score on PC2 reflect molars with relatively quadrangular outline and slight rectangular occlusal polygon more widely displaced to buccally side.

Even though the PCA does not distinguish subfamilies, the plot of PC1 *versus* PC2 (Fig. 2, including 95% confidence ellipses of the subfamily groups) shows clear trends between subfamilies. Alouattinae clearly cluster on the positive scores of PC1, while Pithecinae and Cebinae greatly overlap on the most negative score of PC1. The rest of the groups (Saimirinae, Callicebinae, Callitrichidae, Atellidae, and Aotinae) show intermediate values for PC1 and greatly overlap. For the second PC function (PC2), all groups greatly overlap, though Saimirinae, Callitrichinae and Callicebinae show somewhat higher PC2 scores than the rest. Most of the fossil specimens showed positive PC1 scores, except *Carlocebus* (F5) and especially *Nuciruptor* (F11) and *Cebupithecia* (F12) that had negative PC1 and positive PC2 scores. Most extinct forms overlapped with the extant platyrrhines, within Callicebinae, Callitrichinae, and Atellinae, except *Xenothrix* (F13), *Nuciruptor* and *Cebupithecia*, which do not.

The first two PCs for $M_2$ (Fig. 3) accounted for 42.80% of the total variance (PC1: 28.58%; PC2: 14.22%). The molar shape changes for positive and negative PC1 scores for $M_2$ were relatively similar to those observed for $M_1$, whereas positive PC2 scores for $M_2$ corresponded to the negative ones on PC2 for $M_1$, and negative ones on PC2 for $M_2$ were

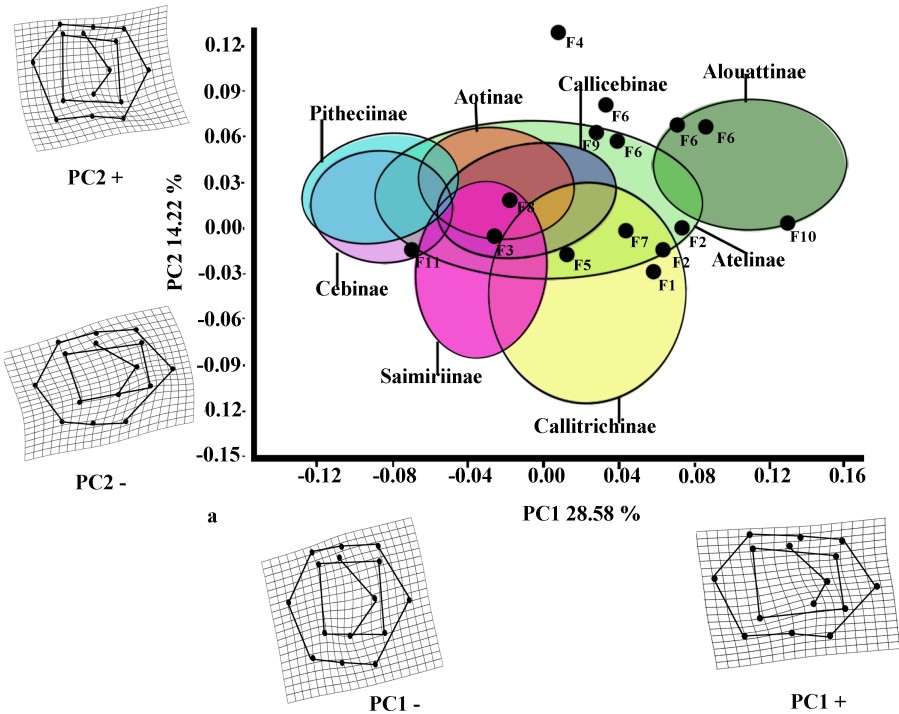

**Figure 3** **Scatterplot of the first two principal components (PCs) derived from the PCA of $M_2$ shape variability of Platyrrhini.** Grids indicate the deformations associated with the extreme values of each principal component. Ellipses represent the subfamily-level classification proposed by *Groves (2005)*. The letters F and numbers in figure represent the fossils listed in Table 1.

equivalent to the positive score of PC2 for $M_1$. The PC1 *versus* PC2 plot (Fig. 3) showed similar distributions of the subfamilies to those for $M_1$, although greater separations between groups were observed. Alouattinae showed the largest, positive scores for PC1, and Pitheciinae and Cebinae the most negative scores, with the other groups showing again intermediate values. Callitrichinae and Saimiriiane were placed mainly on the negative score of the PC2 axis, although they overlapped somewhat with the other groups. Most fossil specimens again clustered on positive scores for PC1 and PC2, mainly within the dispersion of Callitrichinae, although *Stirtonia* (F10), and some specimens of *Neosaimiri* clearly fell within the Alouattinae clade, *Dolichocebus* (F3) within Saimiriinae, and *Nuciruptor* (F11) was closer to Cebinae and Pitheciinae on the negative scores of PC1. *Homunculus* (F4) did not fell at all within any extant taxa, showing highly positive PC2 scores.

## Discriminant analyses of the fossil specimens

The *post-hoc* percentages of correct classification after cross-validation (*pcc*) were high both for $M_1$ (Table 4A, range = [85.7–88.0%]) and $M_2$ (Table 4B, range = [84.7–90.6%]). In both cases the highest *pcc* value was obtained when Groves' *subfamily* factor was discriminated. The range of differences between *pcc* values before and after cross-validation was [1.3–4.7] and in both teeth the *genus* discriminant factor showed the highest decrease in *pcc*. The differences in *pcc* values between Groves' (Cebinae, Saimiriinae, Callitrichinae, Pitheciinae, Callicebinae, Aotinae, Atelinae, Alouattinae) and Rosenberger's (Cebinae, Callitrichinae,

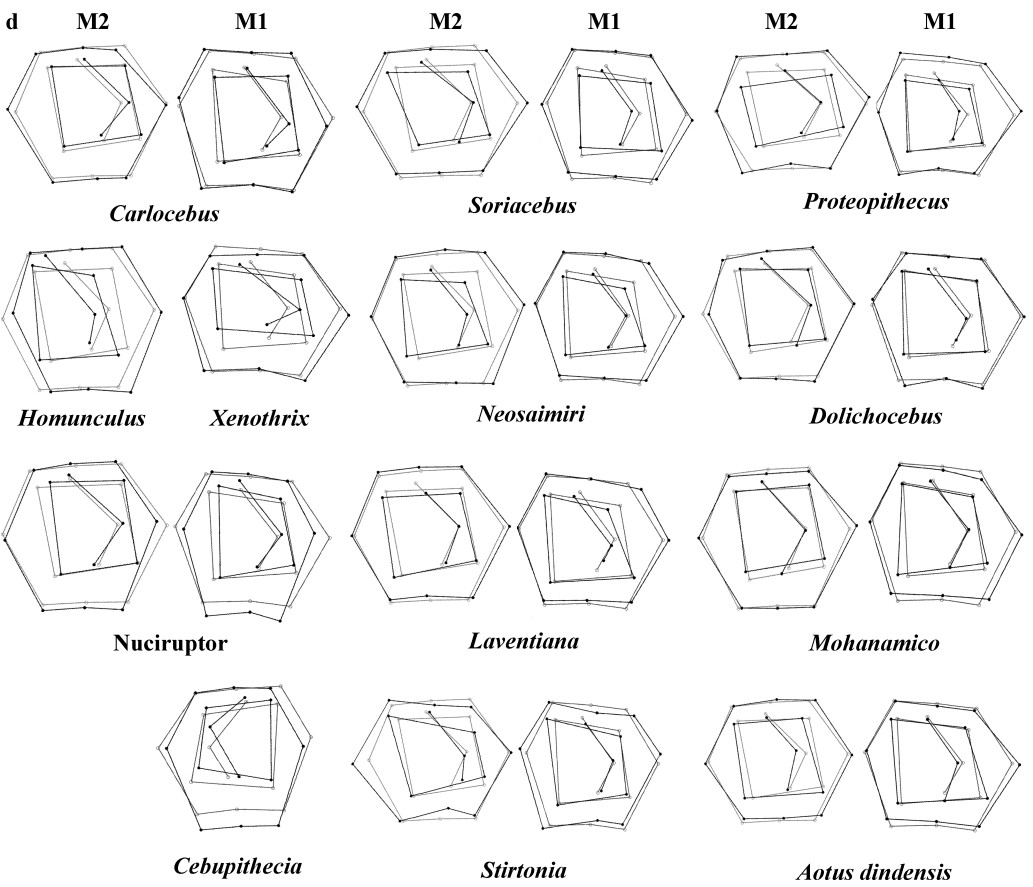

**Figure 4** First and second molar shapes of the extinct fossil platyrhines used in this study.

Pitheciinae, Homunculinae, Atelinae) *pcc* values were 2.3% for $M_1$ and 1.6% for $M_2$ (Table 5). The percentage of total variance explained by the first two discriminant functions (DF1, DF2; Table 4) for all discriminat factors ranged from 63.3% (*genus*) to 100% (*family*) for $M_1$, and from 66.1% (*genus*) to 100% (*family*) for $M_2$. The highest percentage of total variance explained by DF1 was 56.0% (*family*) for $M_1$ and 68.3% (*family*) for $M_2$, and the highest one for DF2 was 44.0% (*family*) for $M_1$ and 32.8% (*subfamily R*) for $M_2$.

Regarding the classification of the fossil specimens, the ranges of the *a priori* classification probabilities varied depending on the discriminant factors used (Table 5; Fig. 4 shows the landmark configurations of the fossil specimes analysed). *Mohanamico* showed a high probability of belonging to the callitrichines clade, as well as *Carlocebus*, although the probability was smaller for $M_2$. Both *Neosaimiri* and *Soriacebus* showed high probabilities of belonging to the callitrichines for $M_1$, though to Callicebinae/Homunculinae for $M_2$. *Cebupithecia* ($M_2$ not available) and *Nuciruptor* neotypes showed a high probability of belonging to the pitheciid clade in LDAs. In contrast, *Xenothrix* ($M_2$ not available) likely belonged to *Callithrix*, despite in the PCA this fossil specimen did not fall within Callitrichinae range. *Stirtonia* was assigned to the Atelidae clade, and to *Alouatta* at the genus level, except for Rosenberger' *subfamily* factor for $M_2$. *Laventiana* was also classified

**Table 5  Summary of the LDA, including the percentage of variance for the two discriminant function (DF1 and DF2), the percentage of original grouped cases correctly classified and the percentage of cross-validated.** Further, the percentage of probability that each case (fossil) belongs to the predicted group. Family: Pitheciidae, Cebidae, Atelidae; subfamily by *Groves (2005)* (Subfamily by G): Aotinae, Cebinae, Saimiriinae, Callitrichinae, Pitheciinae, Callicebinae, Atelinae, Alouattinae; subfamily by *Rosenberger (2011)* (Subfamily by R): Cebinae, Callitrichinae, Pitheciinae, Homunculinae, Atelinae; Genus: The names are listed in Table 2. *Soriacebus*[1,2,3] and *Neosaimiri*[4,5,6,7,8,9,10] correspond to the holotypes numbered on Table 1.

**(A) $M_1$**

|  | Family % | Subfamily by G % | Subfamily by R % | Genus % |
|---|---|---|---|---|
| **DF1** | 56.0 | 50.5 | 42.4 | 49.0 |
| **DF2** | 44.0 | 19.1 | 29.1 | 14.2 |
| **Classification** | 88.7 | 91.3 | 88.2 | 91.0 |
| **Cross-validation** | 87.4 | 88.0 | 85.7 | 86.3 |

| ($M_1$) | Family | % | Subfamily by G | % | Subfamily by R | % | Genus | % |
|---|---|---|---|---|---|---|---|---|
| *Proteopithecus* | Cebidae | 99.6 | Saimiriinae | 99.2 | Cebinae | 99.9 | *Saimiri* | 99.3 |
| *Soriacebus*[1] | Cebidae | 99.9 | Callitrichinae | 99.9 | Callitrichinae | 99.8 | *Saguinus* | 89.6 |
| *Soriacebus*[2] | Cebidae | 99.1 | Callitrichinae | 76.6 | Callitrichinae | 94.0 | *Callithrix* | 69.1 |
| *Dolichocebus* | Cebidae | 86.5 | Callicebinae | 77.9 | Homunculinae | 67.4 | *Callicebus* | 86.4 |
| *Carlocebus* | Cebidae | 97.0 | Callitrichinae | 94.2 | Callitrichinae | 83.7 | *Callithrix* | 87.1 |
| *Neosaimiri*[4] | Pitheciidae | 48.5 | Atelinae | 48.8 | Callitrichinae | 52.2 | *Saguinus* | 78.7 |
| *Neosaimiri*[5] | Cebidae | 98.4 | Callitrichinae | 97.5 | Callitrichinae | 97.3 | *Saguinus* | 99.6 |
| *Neosaimiri*[6] | Cebidae | 97.0 | Callitrichinae | 76.5 | Callitrichinae | 94.6 | *Saguinus* | 72.2 |
| *Laventiana* | Atelidae | 94.6 | Atelinae | 44.5 | Atelinae | 94.9 | *Callicebus* | 53.0 |
| *Mohanamico* | Cebidae | 96.2 | Callitrichinae | 87.3 | Callitrichinae | 70.3 | *Leontopithecus* | 65.4 |
| *Aotus dindensis* | Pitheciidae | 59.0 | Aotinae | 99.7 | Homunculinae | 97.4 | *Aotus* | 98.7 |
| *Stirtonia* | Atelidae | 98.9 | Alouattinae | 99.9 | Atelinae | 98.2 | *Alouatta* | 99.9 |
| *Nuciruptor* | Pitheciidae | 99.7 | Callicebinae | 99.5 | Homunculinae | 83.6 | *Callicebus* | 63.3 |
| *Cebupithecia* | Pitheciidae | 96.5 | Pitheciinae | 92.1 | Pitheciinae | 65.3 | *Chiropotes* | 59.2 |
| *Xenothrix* | Pitheciidae | 75.8 | Callicebinae | 30.5 | Homunculinae | 61.9 | *Callithrix* | 90.7 |

**(B) $M_2$**

|  | Family% | Subfamily by G % | Subfamily by R % | Genus % |
|---|---|---|---|---|
| **DF1** | 68.3 | 45.6 | 47.6 | 43.5 |
| **DF2** | 31.7 | 29.0 | 32.8 | 22.6 |
| **Classification** | 89.5 | 93.3 | 90.3 | 88.7 |
| **Cross-validation** | 88.2 | 90.6 | 89.0 | 84.7 |

| ($M_2$) | Family | % | Subfamily by G | % | Subfamily by R | % | Genus | % |
|---|---|---|---|---|---|---|---|---|
| *Proteopithecus* | Cebidae | 99.4 | Callitrichinae | 82.3 | Callitrichinae | 80.3 | *Callimico* | 86.7 |
| *Soriacebus*[1] | Cebidae | 65.6 | Callicebinae | 81.6 | Homunculinae | 58.4 | *Saguinus* | 74.6 |
| *Soriacebus*[3] | Atelidae | 77.1 | Callitrichinae | 96.7 | Callitrichinae | 98.0 | *Saguinus* | 65.6 |
| *Dolichocebus* | Cebidae | 50.7 | Callicebinae | 92.6 | Homunculinae | 90.1 | *Callicebus* | 92.6 |
| *Homunculus* | Pitheciidae | 91.4 | Callicebinae | 93.7 | Homunculinae | 97.3 | *Callicebus* | 99.9 |
| *Carlocebus* | Cebidae | 55.6 | Callitrichinae | 58.8 | Callitrichinae | 50.4 | *Mico* | 72.5 |
| *Neosaimiri*[7] | Cebidae | 98.3 | Callicebinae | 92.9 | Cebinae | 35.8 | *Callicebus* | 67.2 |

**Table 5** (*continued*)

| ($M_2$) | Family | % | Subfamily by G | % | Subfamily by R | % | Genus | % |
|---|---|---|---|---|---|---|---|---|
| *Neosaimiri*[8] | Cebidae | 64.9 | Callicebinae | 61.2 | Homunculinae | 93.7 | *Saguinus* | 65.1 |
| *Neosaimiri*[9] | Cebidae | 99.5 | Callitrichinae | 61.3 | Callitrichinae | 51.7 | *Saguinus* | 92.3 |
| *Neosaimiri*[10] | Cebidae | 98.9 | Callicebinae | 84.6 | Callitrichinae | 71.9 | *Saguinus* | 98.3 |
| *Laventiana* | Cebidae | 99.9 | Callitrichinae | 99.8 | Callitrichinae | 99.7 | *Saguinus* | 40.8 |
| *Mohanamico* | Cebidae | 97.7 | Callitrichinae | 94.9 | Callitrichinae | 94.6 | *Saguinus* | 99.9 |
| *Aotus dindensis* | Cebidae | 84.4 | Callicebinae | 88.9 | Homunculinae | 76.1 | *Callicebus* | 96.5 |
| *Nuciruptor* | Pithecidae | 89.7 | Pitheciinae | 89.7 | Pitheciinae | 73.0 | *Pithecia* | 49.4 |
| *Stirtonia* | Atelidae | 81.8 | Alouattinae | 86.0 | Callitrichinae | 92.1 | *Alouatta* | 94.0 |

into the atelids for $M_1$, but was more closely related to callitrichines for $M_2$. *Aotus dindensis* showed a high probability of belonging to *Aotus* taxa for $M_1$, but *Callicebus* was the group with the greatest affinity for $M_2$. Finally, *Proteopithecus* showed a high resemblance with *Saimiri* for $M_1$, but with *Callimico* for $M_2$.

## DISCUSSION

The positions of the anthropoid form *Proteopithecus sylviae* (F1) in the morphospace and its molar shapes showed pattern resemblance to that of platyrrhines. However, because, many dental and postcranial features of *P. sylviae* are considered to be symplesiomorphic characters of all anthropoids, it is placed as the stem anthropoid (*Kay, 1990*; *Kay, 2014*). The recent discovery of *Perupithecus ucayaliensis*, probably from the Late Eocene, suggests that this fossil exhibits similarities with *Proteopithecus*, also with *Talahpithecus* and Oligopithecidae (*Bond et al., 2015*). The upper molars of *Perupithecus* slightly resemble those of the callitrichines, but its morphology is more similar to *Proteopithecus* and *Talahpithecus* (*Bond et al., 2015*). *Proteopithecus sylviae* differed from the extant and extinct platyrrhines in having a molar distomesially expanded, marked by a rectangular shape of the occlusal polygon (especially on $M_2$) (also seen in *Xenothrix*). Thus, if the Fayum form likely was a sister taxon to platyrrhines, the interspecific variation of shape would have shown relatively little change. This could mean that the main traits of molars shapes in platyrrhines represent retention of a primitive ancestral form. Moreover, the LDA showed a high probability of *P. sylviae* belonging to the Cebidae clade, suggesting that the molar of the earliest ancestors of platyrrhines must have exhibited close similarity to *Saimiri-Callimico*. This resemblance matches with the description of an Oligocene primate fossil found in South America, *Branisella* (*Rosenberger, 2002*; *Rosenberger et al., 2009*), whose morphology indicates that the structural characteristics of $M_2$ may have been *Saimiri*-like, and the upper $P^2$ *Callimico*-like (*Rosenberger, 1980*). However, both molar shapes of *P. sylviae* more closely resembled *Callimico* than *Saimiri*. Furthermore, the subtriangular upper molars of *Perupithecus*, show relative similarity with *Callimico* (*Bond et al., 2015*). Thus, if *P. sylviae* was a sister taxon of platyrrhines, it is likely that the hypothetical ancestral molar shape of pre-platyrrhine would have been similar to a molar of *Callimico*. By contrast, if *P. sylviae* were a stem species, *Callimico* would show retention of primitive pre-anthropoid platyrrhine molar shape.

## Early Miocene platyrrhines from Patagonia

Most of the traits used to identify phylogenetic affinities among Early Miocene platyrine fossils show high levels of homoplasy *Kay (1990)*, *Kay (2010)* and *Kay (2014)*. The present work alone cannot reject the successive radiations or the long lineages hypotheses, nor can confirm which is correct. However, the PCA showed clear trends at the subfamily level (Figs. 2 and 3). Although the fossils were not very spread out in the morphospace, many of them were located mainly within the Callicebinae and Callitrichinae range (except *Homunculus* for $M_2$), showing phenetic similarities with these two extant subfamilies

The Early Miocene fossils were mainly assigned to two taxa by the LDA; a *Callicebus*-shaped and a *Sagunus*-shaped. For example, *Dolichocebus* (F3) was classified as a pitheciid, mainly by having a square occlusal polygon (Table 4). However, although the PCA for $M_1$ placed this specimen in the Callicebinae range, a morphological similarity with Saimiriinae was seen for $M_2$ (Fig. 3A). In contrast, *Soriacebus* (F2) was related mainly to the callitrichine clade, but for $M_2$ the probability of belonging to this group was small (Table 4). *Soriacebus* showed a rectangular occlusal polygon on $M_2$ and the ectoconid was inclined distolingually. Regarding callitrichines, although *Soriacebus* also showed differences in cusp configuration, the callitrichines and *Soriacebus* share a C-shaped distal side and a somewhat straight lingual-side contour (mostly seen in *Saguinus*). *Kay (1990)* reported that many dental features of marmosets and *Soriacebus* were convergent. In contrast, *Rosenberger, Setoguchi & Shigehara (1990)* suggested that there are some similarities with callitrichines (development of hypoconids and entoconids in the talonid), although, based on the anterior teeth, they concluded that *Soriacebus* represents the first branch of pitheciines. Although marmosets are considered derived lineages (e.g., *Chatterjee et al., 2009*; *Jameson Kiesling et al., 2014*), it is likely that the relation with *Soriacebus* may be due to the fact that callitrichines exhibit primitive traits on their molars, which means that both taxa share a retention of rectangular contour and occlusal polygon shape. In the case of *Carlocebus* (F5), it was classified as a Callitrichinae in the DFA. However, it has been shown to be more similar with *Callicebus* than marmosets, such as the shape contour and quadrate alignment of cusps in both molars. *Homunculus* (F4), was placed outside the range of Patagonian forms in the PCA (Fig. 2A), whereas the LDA indicated a high probability of belonging to Pitheciidae (ca. 91–99%; Table 4), and especially to *Calliecebus*. Nonetheless, *Homunculus* molar showed an asymmetrical shape compared to pitheciid forms. Furthermore, unlike pitheciids, *Homunculus* cusps were predominantly inclined toward the distal side and the trigonid was almost as broad as the basin-like talonid, which means that although sharing some traits with pitheciids, its position is highly uncertain.

## Middle Miocene platyrrhines from Colombia and the Caribbean Xenothrix

Many of these fossils were mostly catalogued as callitrichines, specifically into the *Saguinus* clade, except *Nuciruptor*, *Cebupithecia*, *Aotus dindensis*, and *Stirtonia*. One of the major differences between these primates and the extant forms (except *Alouatta* and *Brachyteles*) was the rectangular-shaped molar (see *Xenothrix* below). This phenetic similarity among phyletically distinct groups of extinct primates indicates that a rectangular-shaped molar
almost certainly represents a plesiomorphy in the Patagonian fossils. Thus, the trend toward ovoid molar shape might be a derived feature in many living forms. *Laventania* (F7) exhibited distally oriented cusps on $M_1$, showing considerable resemblance with some atelid groups, which provided a confusing classification between atelids and *Callicebus* in the LDA (Table 5). Thus, the trend to rectangular shape for $M_1$ in *Laventania* differs notably from the phylogenetic relationship with Cebinae and Saimiriinae. Nonetheless, when $M_2$ was analyzed, the fossil was classified as member of the Callitrichinae clade. As with *Laventania*, some neotypes of *Neosaimiri* (F6) were classified in completely distant taxonomic groups (Table 4). However, despite these results, *Neosaimiri* was principally associated to the Cebidae family, although the molar shape was found to have more affinities with callitrichines than *Saimiri*. On the other hand, *Mohanamico* (F8) and *Aotus dindensis* (F9) have been considered by Kay and collaborators (*Meldrum & Kay, 1997*; *Kay, 2014*) to belong to the same genus, despite *Takai et al. (2009)* have suggested that *A. dindensis* should be assigned to distinct genus. According to their molar shape, *Mohanamico* and *A. dindensis* may be classified into different species. Both fossils showed a relative rectangular shape of the outline, as well as in the occlusal polygon, although $M_2$ in both species was slightly square shaped. In fact, PCA for $M_1$ (Fig. 2A) showed that the two forms were placed closer to each other. Thus, similar molar shape might be due to the fact that the two forms must have shared relatively similar ecological niches, likely because *Mohanamico* and *A. dindensis* were found in the same locality and at the same stratigraphic level (*Kay, 1990*). However, the LDA indicated that the probability of classification was different for both groups. *Aotus dindensis* was mainly related to *Aotus/Callicebus*, whereas *Mohanamico* was assigned to Callitrichinae (Table 4). In the case of *Nuciruptor* (F11) and *Cebupithecia* (F12), the occlusal views in both species were relatively rounded, with a slightly rectangular alignment of cusps, and buccally oriented, which resembles the condition in most extant Pitheciinae. Moreover, the LDA indicated that *Cebupithecia* and *Nuciruptor* had a close affinity with the Pitheciidae clade (Table 4). However, despite the two neotypes clustered close to the pitheciids, they were not placed into the extant species range (except *Nuciruptor* on $M_2$) (Fig. 2A). Several studies have suggested that, although there are important characteristics that have been associated with the living taxa, both fossils should be considered stem pitheciines (*Meldrum & Kay, 1997*; *Kay, Meldrum & Takai, 2013*; *Kay, 2014*).

The sister relationship between *Stirtonia* and *Alouatta* was classified in the LDA with a 99.9% probability for $M_1$ and 94.0% for $M_2$. Likewise, the PCA showed that *Stirtonia* was placed close to howler monkeys (Figs. 2A and 3A). However, differences between *Stirtonia* and *Alouatta* were mainly seen in the occlusal polygon of $M_2$. The metaconid of *Stirtonia* was located near the protoconid and the ectoconid was distolingually inclined, somewhat similar to the *Cebuella* configuration. This relationship was reflected in the high percentage of probability at the subfamily level, Callitrichinae (Table 5).

Finally, *Xenothrix* (F13), the Caribbean platyrrhine form, has been allied with pitheciids (*Rosenberger, 2002*; *Horovitz & MacPhee, 1999*). In the LDA, *Xenothrix* was mainly attributed to pitheciids, but at the genus level, it was assigned to *Callithrix* (Table 4). Thus, some resemblance with marmosets could be interpreted as convergent evolution. However, the relationship between *Xenothrix* and pitheciids was highly uncertain, given

that its molar morphology (especially the occlusal configuration) differs from that of the pitheciids. It is likely that *Xenothrix* could be a single branch that evolved independently of crown platyrrhines, as was suggested by some investigations that proposed an early Antillen arrival (*Iturralde-Vinent & MacPhee, 1999*; *MacPhee & Iturralde-Vinent, 1995*; *MacPhee & Horovitz, 2004*; *Kay et al., 2011*; *Kay, 2014*).

The slow rate of phenotypic changes on molar shapes suggests that morphological stasis (different concept from long lineages hypothesis) explains the low interspecific variation between extinct and extant linages and between Early Miocene platyrrhines (including *P. sylviae*) and forms from La Venta. This small phenotypic variation could be due to developmental and functional constraints, given the role in occlusion and mastication (*Gómez-Robles & Polly, 2012*) and the reduced dietary diversification in platyrrhines. This ecological constraint may be related to the fact that the phenotypic adaptation of main platyrrhine families could have happened in Amazon rainforest (*Jameson Kiesling et al., 2014*). Following an African origin scenario, and taking into account the phenotypic similarity of the most recent discovered and oldest fossil found in Peru, *Perupithecus* (*Bond et al., 2015*), it is likely that the ancestor of extant platyrrhines could have exhibited a *Callimico*-like molar shape. We also observed that *Saguinus* and *Callicebus* were the main assigned groups for Patagonian fossils by the LDA, which suggests that there were both *Callicebus*-like and *Saguinus*-like morphologies in early stem platyrrhines, dispite extant *Saguinus* might not represent an early branch according to molecular evidence. Currently, *Callicebus* and *Saguinus* present relatively high diversity of species and geographic range (*Rylands & Mittermeier, 2009*). The *Callicebus* and *Saguinus* species richness probably are related to expansion and diversification of both clades in the Amazon basin, during the period of platyrrhine evolution (*Ayres & Clutton-Brock, 1992*; *Boubli et al., 2015*). Thus, it is feasible that *Callicebus*, as well as *Saguinus*, molar shape would be an ancestral precursor for the existing forms. Moreover, the Middle Miocene platyrrhines indicate continuity in molar shape pattern with the early fossils, incorporating also new molar shapes not observed in the Patagonian forms: the *Alouatta*-like and the Pitheciinae-like forms.

## CONCLUSIONS

This study develops a dental model based on molar shapes of $M_1$ and $M_2$ to explore phenotypic variation in extinct and extanct platyrrhines. Our results showed that morphological stasis explains the low phenotypic changes in extinct and exctant platyrraines, probably due to ecological constraints, caused by phenotypic adaptation of platyrrhines to a relatively narrow ecological niche. Early and Middle Miocene platyrrhines shared some relatively similar molar shape patterns, whereas the Colombian fossils more closely resemble *Alouatta* and the Pitheciinae. The relation between both fossil samples could be due to: 1. All platyrrhine molar shapes share a primitive retention of the ancestral state. 2. An early divergence between two parallel shapes; a *Callicebus*-like and a *Saguinus*-like, which would be the ancestral precursors to all other forms. 3. *Callicebus*-like and *Saguinus*-like morphologies independently occurred in the early stem platyrrhines.

## ACKNOWLEDGEMENTS

We thank the curators and institutions for allowing us access to specimens and resources: Mario de Vivo and Juliana Gualda Barros (Museu de Zoologia Universidade de São Paulo), Leandro de Oliveira Salles, and we are especially grateful to Sergio Maia Vaz, who supported us with data acquisition (Museu Nacional do Rio de Janeiro). We also thank Mark Teaford and Kenneth Glander for allowing us access to howler monkeys tooth molds from Hacienda La Pacífica (Costa Rica). We also thank Katarzyna Górka for helping in the teeth molding.

### Funding

This work was funded by the Spanish Ministerio de Ciencia e Innovación via projects CGL2011-22999 and SGR2009-884. The funders had no role in study design, data collection and analysis, decision to publish, or preparation of the manuscript.

### Grant Disclosures

The following grant information was disclosed by the authors:
Spanish Ministerio de Ciencia e Innovación: CGL2011-22999, SGR2009-884.

### Competing Interests

The authors declare there are no competing interests.

### Author Contributions

- Mónica Nova Delgado performed the experiments, analyzed the data, wrote the paper, prepared figures and/or tables.
- Jordi Galbany conceived and designed the experiments, reviewed drafts of the paper.
- Alejandro Pérez-Pérez conceived and designed the experiments, contributed reagents/materials/analysis tools, wrote the paper, reviewed drafts of the paper.

### Data Availability

The Geometric morphometric raw and procrustes coordinates of the landmark configurations are available in Dryad (DOI: 10.5061/dryad.3m2g1).

### Supplemental Information

Supplemental information for this article can be found online at http://dx.doi.org/10.7717/peerj.1967#supplemental-information.

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
