# Peer review of "Morphometric variation of extant platyrrhine molars: taxonomic implications for fossil platyrrhines"

_PeerJ, doi:10.7717/peerj.1967_

## Round 0.1 · original submission · Major Revisions

I was able to assemble three reviewers for your manuscript. Their recommendations ranged from Minor to Major Revisions. As you can see one (Reviewer 3) was particularly critical of your manuscript, however all three identified important analytical and interpretive issues that need to be addressed. There is a common concern that the methods used do not directly address the phylogenetic hypotheses. Suggestions include broadening the discussion to include functional interpretations to additional statistical analyses that judge the strength of the phylogenetic signal or account for smaller and various sample sizes. In particular, all three reviewers expressed concern of the appropriateness of the Linear Discriminant Function analysis for the questions at hand. Please address these issue as well as other individual analytical concerns expressed by the reviewers.

There was also a general sense of confusion from the introduction regarding the phylogenetic schemes and fossil sample. Both Reviewer 1 & 2 each suggested additional tables to address these concerns.

Concern was also expressed the different patterns and signals detected for M1 and M2.

Finally, there were many typographical, spelling, and reference errors throughout the manuscript. Many are detailed by Reviewer 2 and the annotated pdfs from Reviewer 3 and me. Please check over the resubmission carefully.

If you feel these concerns can be appropriately addressed, I look forward to receiving a revised manuscript.

·

Basic reporting

This is fine except that the article contained many typos and awkwardly written sentences.

Experimental design

The methods were correctly carried out, however, I do not think that they were entirely appropriate for the research question (see below) and should be supplemented with additional methods that directly test for phylogenetic signal.

Validity of the findings

See above and below.

Additional comments

This paper may be an important contribution to understanding platyrrhine dental diversity, but needs a lot of revision.

First, there are many typos throughout. I marked many in the attached pdf, but I would suggest reading through it very carefully - especially make sure to correctly spell words such as “platyrrhine”, which was misspelled in several different ways. There are also some places where the wording of the sentences is a bit awkward.

Second, and a minor point - the phylogenetic subfamily schemes that you are using are unclear. Perhaps an additional table would be helpful. In reading the paper, you seem to switch back and forth between the scheme that Rosenberger advocates and the one Groves advocates. I just couldn’t follow. Especially be clear about your use of pitheciid, pitheciine, and homunculine.

Third, I remain unconvinced that the methods used are actually giving a strong phylogenetic signal. My own work (Cooke 2011) showed that PC analyses of 3D data returned a dietary signal to a much greater degree than a phylogenetic one. I don’t doubt at all that there is phylogenetic information in the shape data, but how do you separate this from the functional signal? This is especially difficult in a group such as the platyrrhines where dietary adaptations tend to line up along phylogenetic lines. For example, many of the pitheciids are hard object specialists and many of the Callitrichines have a diet high in insects. Take a look at the overlap of pitheciinae and cebinae (fig. 3), for example. They are distantly related according to the genetic data, but their diets have similar structural properties and they overlap quite a bit in shape. So yes, discriminant functions will separate the two into their groups, but for what reasons? Is it phylogenetic relationship, function, both? You may wish to consider measuring the phylogenetic signal in your data using the GeoMorph R package or similar, but you should definitely include some type of phylogenetic methods in addition to the phenetic ones that are employed here. Alternately, the paper could be reframed to be a purely phenetic exploration of shape variation in platyrrhine molars. Either would be fine.

Fourth, now for the fossils, you use discriminant functions to classify them, but what if they are actually stem as Kay advocates? They would be forced into one of the extant platyrrhine families, but maybe they pre-date them. The methods here don’t really account for that possibility, though, you do discuss it briefly in the conclusions.

Fifth, it concerns me that you are getting separate and different signals for the m1 and m2 data. Clearly, there are differences morphologically, but what do these mean? My suspicions are that some of these differences are the result of some of your taxa having a 2.1.3.3 dental formula while others have a 2.1.3.2 formula. There are also major size differences between the two different tooth positions in some of the taxa. While these shape differences are interesting, I think that they are telling you something about function. I would follow up on this more in the discussion. Why is this occurring? In which taxa?

If these issues are addressed, I think it will make a fine contribution.

·

Basic reporting

Nova Delgado et al. present an interesting descriptive study of M1 and M2 shape variation in platyrrhines. They utilize 2D geometric morphometrics to quantify and analyze the among-species variation across their sample, and interpret their results in the context of platyrrhine taxonomy and phylogeny. The work presented here has important descriptive details, but several points need to be addressed before this manuscript can be considered for publication.
Additionally, there are a number of typos and spelling mistakes throughout the manuscript, which need to be corrected. Please check the entire document very carefully, paying particular attention to the species/genera names in the text and tables.
A couple mistakes were also found in the references. Please double-check all the citations before re-submission.

Experimental design

Introduction
Lines 85-142, which describe the different fossil specimens might be better condensed into a Table for clarity.
Methods and Results
Error test: It would be useful to provide comprehensive error tests to establish the reliability of the imaging and subsequent landmarking protocols, especially because slight differences in orientation of the images can lead to substantial measurement error, which can affect the recording of small scale differences in molar shape. Even if this information is provided in earlier publications, it should be referenced and briefly discussed here as well.
LDA: Although LDA has been used in previous publications (Freidline et al., 2012; Harvati, 2003; Gunz and Harvati, 2007; Skinner et al., 2008; Glantz et al., 2009; Mounier et al., 2011; Stansfield and Gunz, 2011) to classify isolated fossils, in a 2D or 3D GM context the number of variables should be less than the number of specimens in each group. This might not be a concern for some of the extant groups in the sample, but it is for the fossils and might inflate the ‘discrimination’ among the groups. To reduce the dimensionality, I suggest conducting the DA using varied numbers of PCs, starting with the first 5-6, going up to 10, to make sure the classifications are not arbitrary. For more details refer to Mitteroecker, P., Gunz, P., 2009. Advances in geometric morphometrics. Evol. Biol. 36, 235e247.

One-way ANOVA results: Fig 2. Shows considerable overlap among most of the groups, except Alouattinae. ANOVA results (p-values) can be affected by sample size (different #of individuals across groups) vs. variables (#landmarks * dimension) and that might be one of the reasons why the ANOVA results are significant between groups. Computing Procrustes distances among the group means might be a better option than ANOVAs.

Allometry (size related shape changes): Even though GPA scales the specimen configuration, removing isometric size, the data still retain allometric variation. It would be interesting to explore this aspect among the groups via a multivariate regression of shape on size.

Validity of the findings

Discussion
It is clear from the PCA plots that M1 and M2 molar morphology shows considerable overlap across the species (with the exception of howler monkeys). The group scatters do not seem to be in keeping with the phylogenetic relationship among these species, but the within-group variation, particularly between the Atelinae and other groups such as Saimiriinae, Callicebinae, is different, suggesting more species-specific changes. It might be worth further discussing how these patterns differ between the two molars, their phylogenetic significance (whether M1 carries a more species-specific signal than M2) and then relating it to Fig. 4.

Additional comments

There are several spelling mistakes and typos through out the text. Here are some, but there are numerous others that need to be addressed.
Minor changes
Line 24: Correct “homoplasia” to “homoplasy”
Lines 24-25: The sentence “Despite dental morphology…” is grammatically incorrect and confusing. Please re-phrase.
Line 25-26: Re-phrase sentence “A geometric morphometric analysis of a 15 landmark-based…” to “A geometric morphometric analysis based on 15 2D landmarks was applied…”
Line 28: Change “Linea” to “Linear”
Line 30: Change “phenipic” to “phenotypic”
Line 31: Correct spelling of platyrrhine.
Line 32: Delete “geometric morphometric” after “reduced”
Line 35: Include “of” between “radiation” and “the main”
Lines 47-48: Sentence construction. Change “Despite” to “Though” or re-phrase entire sentence.
Line 63: Change “is indicative” to “indicates”
Line 82: Please cite the reference, whether Nova Delgado et al 2015a or 2015b is being referenced here.
Line 87: Correct spelling of Caribbean.
Line 127: Remove “1” from “orbits1”
Line 149: Nova Delgado 2014 is not cited in the Bibliography.
Lines 149-151: Please re-phrase to “The aim of the present study is to use 2D GM to quantify and analyze occulsal shape variation of lower molars….”
Lines 156-157: Please provide the relevant references here after “literature”.
Line 205: Change to “used to conduct Linear Discriminant Function analysis”
Line 245: Spelling mistake “higher”
Line 311: Correct spelling of “differs”
Line 323: Correct spelling of “platyrrhines”
Line 339: Correct “may” and “present”
Line 375: Correct spelling of “probabilities”

Reviewer 3 ·

Basic reporting

I realize that the authors are writing in a second language (which is more than I can do), and so I am generally willing to accept small errors here and there. However, there are numerous careless spelling errors throughout the MS (platyrrhine and various Latin taxon names are mis-spelled throughout) that detract from its readability in addition to any expected grammatical errors. Overall, they need to do a better job in the revision.

In terms of the basic structure of the article, the background is very confusing and doesn't do a good job of highlighting the problem. The authors need to present the issue as a problem of taxonomic affinities, but currently the review of fossil platyrrhines in the Introduction appears to make a number of taxonomic decisions that are, in fact, contested. The authors need to simply make the point that the taxonomic and phylogenetic positions of many early fossil platyrrhines are disputed, and so they are attempting an analysis of molar shape to see if it helps clarify anything or support one hypothesis or another. Right now, it reads like a confusing review of mostly Rosenberger's preferred taxonomy.

Figures are fine.

Experimental design

As mentioned above, the experimental design needs to be better articulated and set-up in the Introduction. As you will see in my annotated PDF, I had concerns with the methodology as well. First, taking published photographs for all fossil specimens seems to introduce potential error because the published photos are, of course, not taken using the same methodology as the extant sample. I would encourage the authors to get a hold of casts of the fossil specimens at the very least, if not photograph the original fossils themselves, in order to standardize their sample as much as possible. In addition, only 1 specimen represents each fossil taxon, even though some of the taxa included have decent size hypodigms. A better effort should be made to expand the sample if possible. Finally, running a LDA is a bit counter-intuitive for the question at hand.....since one of the major hypotheses out there (i.e., Kay and colleagues) suggests that most of the early Patagonian fossil taxa do not have any relationship to the extant taxa, running a LDA and forcing the fossils to be placed in extant groups is totally uninformative. IN other words, by including the fossils in the LDA, you have already made the assumption that they can be assigned to extant taxa, but one of the hypotheses you are trying to test does not accept this assumption to begin with! The authors need to re-think their analyses a bit if they really want to get at the question of testing long-lineage vs. layered hypothesis. The same problem exists for including Proteopithecus in the LDA since most authorities do not consider it a crown platyrrhine.

Methods should be clearer on what exactly is being examined its the ANOVAs and the Results need to be clearer on what features are loading on each of the principal components, particularly if size is included on any of them.

Validity of the findings

Data are, for the most part, fine. However, given the small sample sizes involved and the methodological issues noted above and on the annotated PDF, I don't think the study really adds anything to our understanding of platyrrhine evolution. They present an interesting idea in the Discussion about morphological stasis in platyrrhines, but they have no comparative data to be able to make the claim that platyrrhines show less interspecific variation compared to other primate groups such as cercopithecoids or hominoids. In order to make these claims, additional data and analyses must be performed or referenced somewhere.

Additional comments

Overall, the study is a nice start, but I think significant revisions are necessary to make this a contribution that adds something to the literature. As it stands, the results don't add anything to the debate and don't help clarify fossil taxonomy or phylogeny at all. The fossil sample needs to be expanded and the methods need to be more carefully thought out.....the LDA does not really get at the question they are interested in. Please see my annotated PDF for further comments/suggested edits.

Annotated reviews are not available for download in order to protect the identity of reviewers who chose to remain anonymous.

---

## Round 0.2 · Minor Revisions

Thank you for submitting your revised manuscript. I sought the opinion of two of the previous reviewers, and they both agree, as do I, that your manuscript is substantially improved. They both have a few questions and suggested edits. Note that the one of the reviewers has provided an annotated pdf of your manuscript with specific suggestions. If you can suitably address these concerns, I look forward to receiving your revised manuscript. Please be sure to check grammar and spelling issues closely, because articles that are accepted move to the production stage quickly.

·

Basic reporting

The article is much improved over the last submission. I have made some comments in the attached pdf.

Experimental design

The experimental design is good.

Validity of the findings

The findings are valid. I do recommend some softening of the language in areas where the authors draw phylogenetic conclusions without performing phylogenetic analyses. On the whole, their findings support phenetic relationships between certain groups.

Additional comments

This is a reasonable revision, and you adequately addressed my concerns. Please the attached pdf for more detailed comments. It was a pleasure reading your paper.

·

Basic reporting

The paper is much improved, but there are still a few points that need clarification. It would also help to edit some of the longer sentences for clarity (introduction and discussion sections) and pay attention to the grammar.

Experimental design

I am still concerned about the methodology and data obtained from photographs. Even though the protocol has been published elsewhere, any data obtained from photographs need to be backed-up by rigorous error tests, which are lacking in this paper and elsewhere. An error test involves measuring the same landmarks in the same scale and same orientation on a set number of specimens multiple times to ensure minimum observer error (inter and intra). If such a test was not performed, I strongly urge the authors to do so. Or if an error test was performed in the previous publications then it would be helpful to clearly state that or briefly provide the results of such tests.

In addition, the sentence provided (line 179) is grammatically incorrect.

Note on DFA:
To clarify, the suggestion made previously was not to use DFA to show shape changes (that is not possible and it is not what a DFA is used for) nor was it to question the objective of a PCA. The suggestion was to do the DFA on the PC scores, for example taking the scores of PC1 to PC10. The authors do state they used “PCs” (Line 202) to run the DFA; I'm assuming they meant the scores. Please be clearer here so as to not confuse the readers.

However, I am still not sure why the authors used a DFA at all in the paper and how it relates to their question.

Validity of the findings

Please see above comments.

Additional comments

No comments

---

## Round 0.3 · accepted · Accept

I am glad to see that you have addressed the reviewers' concerns. This is a very interesting study. In my last read through I caught a few more grammatical/spelling errors. I've included them as a list below. However, as the manuscript goes straight to production, you will have to address these in the proof if they aren't caught by the copy editor.

Abstract: Line 30 – Patternss

Line 45: group ,

Line 107: “.e.g” should be “e.g.,”

Line 120: Delete “traits”

Line 132: Add “suggesting that IT is likely”

Line 162: Delete the s at the end of “haplotypes”

Line 202: “Further” to “Furthermore”

Line 326: “explaine” to “explained”

Line 350: Figure 4 legend. “Firts”

Line 373: Delete “a” between P2 and Callimico

Line 449: “excep” to “except”

Line 505: “development” to “developmental”

Line 542: “PlatyrrhineS”

Lines 542-544: Maybe rephrase to “probably due to ecological constraints caused by phenotypic adaptation of plattyrrhines to a relatively narrow ecological niche.”

Lines 544-546: The phrasing of this sentence is unclear. Please revise.

Line 549: Possibly rephrase to “Callicebus-like and Saguinus-like morphologies independently occurring in the early stem platyrrhines.” ?

Line 557: “allowimg”